# STATIC PREDICTION OF RUNTIME ERRORS BY LEARNING TO EXECUTE PROGRAMS WITH EXTERNAL RESOURCE DESCRIPTIONS

**David Bieber**
Google Research

**Rishab Goel**
Mila

**Daniel Zheng**
Google Research

**Daniel Tarlow**
Google Research

**Hugo Larochelle**
Google Research

## ABSTRACT

The execution behavior of a program often depends on external resources, such as program inputs or file contents, and so the program cannot be run in isolation. Nevertheless, software developers benefit from fast iteration loops where automated tools identify errors as early as possible, even before programs can be compiled and run. This presents an interesting machine learning challenge: can we predict runtime errors in a "static" setting, where program execution is not possible? Here, we introduce a competitive programming dataset and task for predicting runtime errors, which we show is difficult for generic models like Transformers. We approach this task by developing an interpreter-inspired architecture with an inductive bias towards mimicking program executions, which models exception handling and "learns to execute" descriptions of external resources. Surprisingly, we show that the model can also predict the locations of errors, despite being trained only on labels indicating error presence or absence and kind. In total, we present a practical and difficult-yet-approachable challenge problem related to learning program execution behavior and we demonstrate promising new capabilities of interpreter-inspired machine learning models for code.

## 1 INTRODUCTION

We investigate applying neural machine learning methods to the static analysis of source code for early prediction of runtime errors. The execution behavior of a program is in general not fully defined by its source code in isolation, because programs often rely on external resources like inputs, the contents of files, or the network. Nevertheless, software developers benefit from fast iteration loops where automated tools identify errors early, even when program execution is not yet an option. Therefore we consider the following machine learning challenge: can we predict runtime errors in a "static" setting, where program execution is not possible?

This runtime error prediction task is well suited as a challenge problem because it is difficult-yet-approachable, has real-world value for software developers, requires novel modeling considerations that we hypothesize will be applicable to a range of learning for code tasks, and with this work, now has a suitable large dataset of complex human-authored code with error labels. The task is to predict whether a program will exhibit a runtime error when it is run, and if so to determine the error; even when static analysis cannot provide guarantees of an error in the code, patterns learned from data may point to likely errors. Our dataset consists of 2.4 million Python 3 programs from Project CodeNet (Puri et al., 2021) written by competitive programmers. We have run all programs in a sandboxed environment on sample inputs to determine their error classes, finding the programs exhibit 26 distinct error classes including "no error". Each program relies on an external resource, the stdin input stream, and we pair each program with a natural language description of the behavior of the stream. We make the task and dataset, along with all models considered in this work, available for the research community to facilitate reproduction of this work and further research[1].

To make progress on this challenging task, we identify a promising class of models from prior work, interpreter-inspired models, and we demonstrate they perform well on the task. Instruction Pointer

---

[1]https://github.com/google-research/runtime-error-prediction

Attention Graph Neural Network (IPA-GNN) (Bieber et al., 2020) models simulate the execution of a program, following its control flow structure, but operating in a continuous embedding space. We make a number of improvements to IPA-GNN: scaling up to handle complex programs requiring thousands of execution steps, adding the ability to "learn to execute" descriptions of external resources, and extending the architecture to model exception handling and recover error locations. We evaluate these interpreter-inspired architectures against Transformer, LSTM, and GGNN neural baselines, and against pylint as a static analysis baseline. Our combined improvements lead to increased accuracy in predicting runtime errors and to interpretability allowing for prediction of error locations even though the models are only trained on error presence and error class, not error location. In total, we summarize our contributions as:

- We introduce the runtime error prediction task and a large accompanying dataset, providing runtime error annotations for millions of competition Python programs.
- We demonstrate that IPA-GNN architectures are practical for the complexity of real programs by scaling them to handle competition programs, and there we find they outperform generic models.
- We demonstrate that external resource descriptions, such as Japanese or English descriptions of stdin, can be leveraged to improve performance on the task across all model architectures.
- We extend the IPA-GNN to model exception handling, resulting in the Exception IPA-GNN, which we find can localize errors even when only trained on error presence and kind, not error location.

## 2    RELATED WORK

**Program analysis**    Program analysis is a rich family of techniques for detecting defects in programs, including static analyses which are performed without executing code (Livshits and Lam, 2005; Xie and Aiken, 2006; Ayewah et al., 2008) and dynamic analyses which are performed at runtime (Cadar et al., 2008; Sen et al., 2005; Godefroid et al., 2005). Linters and type checkers are popular error detection tools that use static analysis. Static analysis (e.g. symbolic execution) does not typically use concrete inputs, while dynamic analysis requires concrete inputs and program execution. Compared with traditional static analysis, our approach is more flexible in its input representation, using a general "resource description" abstraction, which can represent the entire spectrum from concrete inputs to input constraints to missing inputs.

**Execution-aware models**    Several neural architectures draw inspiration from program interpreters (Graves et al., 2014; Łukasz Kaiser and Sutskever, 2016; Reed and de Freitas, 2016; Graves et al., 2016; Bošnjak et al., 2017; Gaunt et al., 2017; Dehghani et al., 2019; Bieber et al., 2020). Our work is most similar to Bieber et al. (2020) and Bošnjak et al. (2017), focusing on how interpreters handle control flow and exception handling, rather than on memory allocation and function call stacks. Other works use program execution data directly, training with textual representations of execution traces as inputs (Nye et al., 2021a; Pei et al., 2021; Nye et al., 2021b) or performing execution during synthesis (Chen et al., 2019; Li et al., 2022; Shrivastava et al., 2021). Compared with these, our approach uses weaker supervision, using only runtime error labels for training.

**Fault detection and localization datasets**    There has been considerable recent interest in applying machine learning to identifying and localizing faults in source code (Allamanis et al., 2018a). Puri et al. (2021) makes a large dataset of real world programs available, which we build on in constructing our runtime errors dataset. Our dataset (i) is large (it has millions of examples), (ii) exhibits many programming language features, (iii) is written by human authors, and (iv) has error labels from the execution behavior of programs. Previous code datasets only exhibit a subset of these properties: large real-world and competition code datasets (Hendrycks et al., 2021; Li et al., 2022; Kanade et al., 2020; Raychev et al., 2016; Husain et al., 2019; Puri et al., 2021) exhibit properties i, ii, and iii, but not iv, while learning to execute datasets (Zaremba and Sutskever, 2014; Bieber et al., 2020) exhibit property iv but not i, ii, or iii. Recent program synthesis datasets (Chen et al., 2021; Austin et al., 2021) exhibit ii and iii only. Other datasets obtain error labels by injecting synthetic errors (Allamanis et al., 2018b; Karampatsis and Sutton, 2020; Pradel and Sen, 2018) (lacking the realism of iii) or from commit messages (Just et al., 2014; Dinella et al., 2020) (lacking i and iv).

**Fault localization approaches**    Fault localization approaches vary in (i) level of supervision – weak (error labels) (Li et al., 2019) vs strong (explicit location labels) (Lou et al., 2021; Zhang et al.,

| TARGET CLASS | TRAIN # | VALID # | TEST # | TARGET CLASS | TRAIN # | VALID # | TEST # |
|---|---|---|---|---|---|---|---|
| | | | | numpy.AxisError | 20 | 2 | 3 |
| No error | 1881303 | 207162 | 205343 / 13289† | OSError | 19 | 2 | 2 |
| | | | | OverflowError | 62 | 6 | 11 |
| AssertionError | 47 | 4 | 8 | re.error | 5 | 0 | 0 |
| AttributeError | 10026 | 509 | 1674 | RecursionError | 2 | 0 | 1 |
| EOFError | 7676 | 727 | 797 | RuntimeError | 24 | 5 | 3 |
| FileNotFoundError | 259 | 37 | 22 | StopIteration | 3 | 0 | 1 |
| ImportError | 7645 | 285 | 841 | SyntaxError | 74 | 4 | 3 |
| IndentationError | 10 | 0 | 12 | TypeError | 21414 | 2641 | 2603 |
| IndexError | 7505 | 965 | 733 | UnboundLocalError | 8585 | 991 | 833 |
| KeyError | 362 | 39 | 22 | ValueError | 25087 | 3087 | 2828 |
| MemoryError | 8 | 7 | 1 | ZeroDivisionError | 437 | 47 | 125 |
| ModuleNotFoundError | 1876 | 186 | 110 | Timeout | 7816 | 1072 | 691 |
| NameError | 21540 | 2427 | 2422 | Other | 18 | 8 | 2 |

Table 1: Distribution of target classes in the runtime errors dataset. † denotes the balanced test split.

2019; Zhou et al., 2019; Allamanis et al., 2021) – and (ii) localization granularity – statement-level (Lou et al., 2021; Zhang et al., 2019; Allamanis et al., 2021) vs method-level (Li et al., 2019; Zhou et al., 2019). Our approach uses weak supervision in the form of runtime error labels to indirectly learn fault localization at a statement-level.

## 3 RUNTIME ERROR PREDICTION

**Task** The goal of the runtime error prediction task is to determine statically whether a program is liable to encounter a runtime error when run, and if so, what error kind. The programs cannot be executed directly, as they lack unit tests and depend on external resources which are not available. A textual description of the external resources, which may be the program's inputs, a file's contents, or network access, is provided. This makes reasoning about the execution behavior of the programs plausible, even though actually performing the execution is not. We treat this task as one-class classification, with each error type as its own class and with "no error" as an additional class.

**Dataset** We construct the runtime errors dataset using Python submissions to competitive programming problems from Project CodeNet (Puri et al., 2021). Beginning with the 3.28 million Python submissions in Project CodeNet, we filter the submissions to keep only those written in Python 3, which are syntactically valid, which do not make calls to user-defined functions, and which do not exceed a threshold length of 512 tokens once tokenized. By running each submission in a sandboxed environment, we identify its ground truth runtime error class. Each submission is associated with a competitive programming problem whose problem statement we parse to obtain a description in either English or Japanese of the inputs the program is liable to receive at runtime. This process results in a dataset of 2.44 million submissions, each paired with one of 26 target classes. The "no error" target is most common, accounting for 93.4% of examples. For examples with one of the other 25 error classes, we additionally note the line number at which the error occurs.

We divide the problems into train, validation, and test splits at a ratio of 80:10:10. All submissions to the same problem become part of the same split. This reduces similarities between examples across splits that otherwise could arise from the presence of multiple similar submissions for the same problem. Since there is a strong class imbalance in favor of the no error class, we also produce a balanced version of the test split by sampling the no error examples such that they comprise roughly 50% of the test split. We use this balanced test split for all evaluations. We report the number of examples having each target class in each split in Table 1. We describe the full dataset generation and filtering process in greater detail in Appendix A, and we evaluate the limitations of the dataset quantitatively in Appendix B.

While there are many datasets in the literature that test understanding of different aspects of code including bug-finding, we believe ours fills a gap: it is large-scale (millions of examples), it has real-world implications and presents a practical opportunity for improvement using ML-based approaches, and it tests a combination of statistical reasoning and reasoning about program execution.

---

**Algorithm 1**   Interpreter for which the Exception IPA-GNN is a continuous relaxation

---

**Input:** Program $x$
1: $h \leftarrow \varnothing; p \leftarrow 0$                                         ▷ Initialize the interpreter.
2: **while** $p \notin \{n_{\text{exit}}, n_{\text{error}}\}$ **do**
3:     $h \leftarrow \text{Evaluate}(x_p, h)$                             ▷ Evaluate the current statement.
4:     **if** $\text{Raises}(x_p, h)$ **then**
5:         $p \leftarrow \text{GetRaiseNode}(x_p, h)$                              ▷ Raise exception.
6:     **else**
7:     **if** $\text{Branches}(x_p, h)$ **then**
8:         $p \leftarrow \text{GetBranchNode}(x_p, h)$                                 ▷ Follow branch.
9:     **else**
10:         $p \leftarrow p + 1$                                    ▷ Proceed to next statement.

---

# 4   APPROACH: IPA-GNNS AS RELAXATIONS OF INTERPRETERS

We make three modifications to the Instruction Pointer Attention Graph Neural Network (IPA-GNN) architecture. These modifications scale the IPA-GNN to complex code, allow it to incorporate external resource descriptions into its learned executions, and add support for modeling exception handling. The IPA-GNN architecture is a continuous relaxation of the standard interpreter ($I$) defined by the pseudocode in Algorithm 1, minus the magenta text. We frame these modifications in relation to specific lines of the algorithm: scaling the IPA-GNN to complex human-authored code (Section 4.1) and incorporating external resource descriptions (Section 4.2) both pertain to interpreting and executing statement $x_p$ at Line 3, and modeling exception handling adds the magenta text at lines 4-6 to yield a new interpreter ($I'$) (Section 4.3). We showcase the behavior of both interpreters $I$ and $I'$ on a sample program in Figure 1, and illustrate an execution of the same program by a continuous relaxation of interpreter $I'$ ($\tilde{I}'$) alongside it.

## 4.1   EXTENDING THE IPA-GNN TO REAL PROGRAMS

Bieber et al. (2020) interprets the IPA-GNN architecture as a message passing graph neural network operating on the statement-level control-flow graph of the input program $x$. Each node in the graph corresponds to a single statement in the program. At each step $t$ of the architecture, each node performs three steps: it executes the statement at that node (Line 3, Equation 2), computes a branch decision (Lines 7-8, Equation 4), and performs mean-field averaging over the resulting states and instruction pointers (Appendix C, Equations 10 and 11).

Unlike in Bieber et al. (2020) where program statements are simple enough to be uniformly encoded as four-tuples, the programs in our runtime errors dataset consist of arbitrarily complex Python

| $n$ | SOURCE |
|---|---|
| 1 | `x = input()` |
| 2 | `if x > 0:` |
| 3 | `    y = 4/3 * x` |
| 4 | `else:` |
| 5 | `    y = abs(x)` |
| 6 | `z = y + sqrt(x)` |
| 7 | `<exit>` |
| 8 | `<raise>` |

(a) A sample program illustrative of Algorithm 1 behavior, which raises a ValueError if $x < 0$ at line 6.

| | STDIN | $-3$ |
|---|---|---|
| | STDIN DESCRIPTION | `"A SINGLE INTEGER -10..10"` |

(b) The resource description suggests values the program may receive on stdin.

| $t$ | $h_{A,B}$ | $p_I$ | $p_{I'}$ | $h_{\tilde{I}'}$ | $p_{\tilde{I}'}$ |
|---|---|---|---|---|---|
| 0 | {} | 1 | 1 | ⬜ | 10000000 |
| 1 | {x: -3} | 2 | 2 | 🔳 | 01000000 |
| 2 | {x: -3} | 5 | 5 | 🔳 | 00001000 |
| 3 | {x: -3, y: 3} | 6 | 6 | 🟦 | 00000100 |
| 4 | ValueError: line 6 | 7 | 8 | 🟨 | 00000001 |

(c) Step-by-step execution of the program under interpreters $I$ and $I'$, and continuous relaxation $\tilde{I}'$. Distinct colors represent distinct embedding values.

Figure 1: A sample program and its execution under discrete interpreters $I$ and $I'$ (Algorithm 1) and under continuous relaxation $\tilde{I}'$ of interpreter $I'$. $p_{I_t}$ denotes the instruction pointer under $I$ at step $t$.

statements authored by real programmers in a competition setting. The language features used are numerous and varied, and so the statement lengths vary substantially, with a mean statement length of 6.7 tokens; we report the full distribution of statement lengths in Figure 4.

The IPA-GNN architecture operates on a program $x$'s statement-level control-flow graph, and so requires per-statement embeddings $\text{Embed}(x_n)$ for each statement $x_n$. We first apply either a *local* or *global* Transformer encoder to produce per-token embeddings, and we subsequently apply one of four pooling variants to a span of such embeddings to produce a *node embedding* per statement in a program. In the local approach, we apply an attention mask to limit the embedding of a token in a statement to attending to other tokens in the same statement. In the global approach, no such attention mask is applied, and so every token may attend to every other token in the program. We consider four types of pooling in our hyperparameter search space: *first*, *sum*, *mean*, and *max*. The resulting embedding is given by

$$\text{Embed}(x_n) = \text{Pool}\big(\text{Transformer}(x)_{\text{Span}(x,n)}\big). \tag{1}$$

First pooling takes the embedding of the first token in the span of node $n$. Sum, mean, and max pooling apply their respective operations to the embeddings of all tokens in the span of node $n$.

Finally we find that the programs in our dataset require as many as 174 steps of the IPA-GNN under the model's heuristic for step limit $T(x)$ (Appendix E). To reduce the memory requirements, we apply rematerialization at each layer of the model (Griewank and Walther, 2000; Chen et al., 2016).

## 4.2 Executing with Resource Descriptions

In our dataset, each program $x$ may be accompanied by a description of what values stdin may contain at runtime. We convert this description into embedding $d(x)$; the embeddings, vocabulary, and tokenizer used to produce $d(x)$ are shared with those used to produce token embeddings from program source. Analogous to Line 1 of Algorithm 1, IPA-GNN architectures initialize with per-node hidden states $h_{0,:} = 0$ and soft instruction pointer $p_{0,n} = \mathbb{1}\{n = 0\}$. Here $p_{t,n}$ represents the probability under the model that node $n$ is executing at step $t$. Following initialization, each step of an IPA-GNN begins by simulating execution (Line 3) of each non-terminal statement with non-zero probability under the soft instruction pointer to propose a new hidden state contribution

$$a_{t,n}^{(1)} = \text{RNN}(h_{t-1,n}, \text{Modulate}(\text{Embed}(x_n), d(x), h_{t-1,n})). \tag{2}$$

The text in magenta shows our modification to the IPA-GNN architecture to incorporate external resource descriptions. We consider both *Feature-wise Linear Modulation* (FiLM) (Perez et al., 2017) and *cross-attention* (Lee et al., 2019) for the $\text{Modulate}$ function, which we define in Appendix D. Modulation allows the IPA-GNN to execute differently at each step conditioned on the information in the resource description, whether it be type information, value ranges, or candidate values.

We also consider an additional method: injecting the description as a *docstring* at the start of the program. This method yields a new valid Python program, and so any model can accommodate it.

## 4.3 Modeling Exception Handling

The final modification we make to the IPA-GNN architecture is to model exception handling. In Algorithm 1, this corresponds to adding the magenta text to form interpreter $I'$, computing a raise decision (Lines 4-6, Equation 3). We call the architecture that results the Exception IPA-GNN.

Whereas execution always proceeds from statement to next statement in interpreter $I$ and in the IPA-GNN, interpreter $I'$ admits another behavior. Under $I'$ and the Exception IPA-GNN, execution may proceed from any statement to a surrounding "except block", if it is contained in a try/except frame, or else to a special global error node, which we denote $n_{\text{error}}$. In the sample execution in Figure 1c we see at step $t = 4$ the instruction pointer $p_{I'}$ updates to $n_{\text{error}} = 8$.

We write that the IPA-GNN makes raise decisions as

$$b_{t,n,r(n)}, (1 - b_{t,n,r(n)}) = \text{softmax}\left(\text{Dense}(a_{t,n}^{(1)})\right). \tag{3}$$

The dense layer here has two outputs representing the cases that an error is and is not raised. Here $r(n)$ denotes the node that statement $n$ raises to; $r(n) = n_{\text{error}}$ if $n$ is not contained in a try/except frame, and $b_{t,n,n'}$ denotes the probability under the model of execution transitioning from $n$ to $n'$.

Next the model makes soft branch decisions in an analogous manner; the dense layer for making branch decisions has distinct weights from the layer for making raise decisions.

$$b_{t,n,n_1}, b_{t,n,n_2} = (1 - b_{t,n,r(n)}) \cdot \text{softmax}\left(\text{Dense}(a_{t,n}^{(1)})\right). \tag{4}$$

The text in magenta corresponds to the "else" at Line 6. The model has now assigned probability to up to three possible outcomes for each node: the probability that $n$ raises an exception $b_{t,n,r(n)}$, the probability that the true branch is followed $b_{t,n,n_1}$, and the probability that the false branch is followed $b_{t,n,n_2}$. In the common case where a node is not a control node and has only a single successor, the probability of reaching that successor is simply $1 - b_{t,n,r(n)}$.

Finally, we assign each program a step limit $T(x)$ using the same heuristic as Bieber et al. (2020), detailed in Appendix E. After $T(x)$ steps of the architecture, the model directly uses the probability mass at $n_{\text{exit}}$ and $n_{\text{error}}$ to predict whether the program raises an error, and if so it predicts the error type using the hidden state at the error node. We write the modified IPA-GNN's predictions as

$$P(\text{no error}) \propto p_{T(x),n_{\text{exit}}} \text{ and } P(\text{error}) \propto p_{T(x),n_{\text{error}}}, \text{ with} \tag{5}$$

$$P(\text{error} = k \mid \text{error}) = \text{softmax}\left(\text{Dense}(h_{T(x),n_{\text{error}}})\right). \tag{6}$$

We train with a cross-entropy loss on the class predictions, treating "no error" as its own class.

### 4.4 Unsupervised Localization of Errors

Since the Exception IPA-GNN makes soft decisions as to when to raise an exception, we aggregate these soft decisions to obtain the model's prediction for where a program raises an error. We use this to evaluate the model's localization accuracy despite training without error locations as supervision.

For programs that lack try/except frames, we compute the localization predictions of the model by summing, separately for each node, the contributions from that node to the error node across all time steps. This gives an estimate of *exception provenance* as

$$p(\text{error at statement } n) = \sum_t p_{t,n} \cdot b_{t,n,n_{\text{error}}}. \tag{7}$$

For programs with a try/except frame, we must trace the exception back to the statement that originally raised it. To do this we calculate a recurrence as detailed in Appendix H.

## 5 Experiments

In our experiments we evaluate the following research questions:

**RQ1:** How does the adaptation of the IPA-GNN to realistic code compare against existing static analysis and against standard architectures like GGNN, LSTM, and Transformer? (Section 5.1)

**RQ2:** What is the impact of including resource descriptions? What methods for incorporating them work best? (Section 5.2)

**RQ3:** How interpretable are the soft instruction pointer values in the Exception IPA-GNN for localizing errors? How does unsupervised localization with the Exception IPA-GNN compare to alternative unsupervised localization approaches based on multiple instance learning? (Section 5.3)

### 5.1 Evaluation of IPA-GNN Against Baselines

We describe the experimental setup for our first experiment, comparing the IPA-GNN architectures with Transformer (Vaswani et al., 2017), GGNN (Li et al., 2017), and LSTM (Hochreiter and Schmidhuber, 1997) baselines. In all approaches, we use the 30,000 token vocabulary constructed in Appendix A, applying Byte-Pair Encoding (BPE) tokenization (Sennrich et al., 2016) to tokenize each program into a sequence of token indices. The Transformer operates on this sequence of token indices directly, with its final representation computed via mean pooling. For all other models (GGNN, LSTM, IPA-GNN, and Exception IPA-GNN), the token indices are first combined via a masked (local) Transformer to produce per-node embeddings, and the model operates on these per-node embeddings as in Section 4.1. Following Bieber et al. (2020) we encode programs for a GGNN using six edge types, and use a two-layer LSTM for the LSTM baseline and in all IPA-GNN variants.

(a) Accuracy, weighted F1, and weighted error F1 scores.

| | MODEL | R.D.? | ACC. | W. F1 | E. F1 |
|---|---|---|---|---|---|
| BASE-LINES | PYLINT | | 60.4 | 47.9 | 23.8 |
| | GGNN | | 62.8 | 58.9 | 45.8 |
| | TRANSFORMER | | 63.6 | 60.4 | 48.1 |
| | LSTM | | 66.1 | 61.4 | 48.4 |
| ABLATIONS | GGNN | ✔ | 68.3 | 66.5 | 56.8 |
| | TRANSFORMER | ✔ | 67.3 | 65.1 | 54.7 |
| | LSTM | ✔ | 68.1 | 66.8 | 58.3 |
| | IPA-GNN | | 68.3 | 64.8 | 53.8 |
| | E. IPA-GNN | | 68.7 | 64.9 | 53.3 |
| OURS | IPA-GNN | ✔ | 71.4 | 70.1 | 62.2 |
| | E. IPA-GNN | ✔ | **71.6** | **70.9** | **63.5** |

(b) Localization accuracy (%) for the MIL Transformers and Exception IPA-GNN.

| MODEL | R.D.? | LOCAL. |
|---|---|---|
| LOCAL MIL TRANSFORMER | | 33.0 |
| LOCAL MIL TRANSFORMER | ✓ | 48.9 |
| GLOBAL MIL TRANSFORMER | | 48.2 |
| GLOBAL MIL TRANSFORMER | ✓ | 48.8 |
| E. IPA-GNN | | 50.8 |
| E. IPA-GNN + DOCSTRING | ✓ | 64.7 |
| E. IPA-GNN + FiLM | ✓ | 64.5 |
| E. IPA-GNN + CROSS ATTENTION | ✓ | **68.8** |

Table 2: Error classification and error localization results on the balanced test set with and without resource descriptions (R.D.).

In order to compare against the capabilities of a standard static analysis setup, we also consider a baseline based on pylint. For this baseline, we map a subset of the findings that pylint can identify to runtime error classes that they can indicate. The baseline predicts an error class if pylint identifies a corresponding finding. The purpose of this baseline is to consider a standard tool used by Python developers and see how it is performing on the task. We provide further details in Appendix G.

For each neural approach, we perform an independent hyperparameter search using random search. We list the hyperparameter space considered and model selection criteria in Appendix E. The models are each trained to minimize a cross-entropy loss on the target class using stochastic gradient descent for up to 500,000 steps with a mini-batch size of 32. In order to more closely match the target class distribution found in the balanced test set, we sample mini-batches such that the proportion of examples with target "no error" and those with an error target is 1:1 in expectation. We evaluate the selected models on the balanced test set, and report the results in Table 2a (see rows without check marks). Weighted F1 score (W. F1) performs a weighted average of the per-class F1 scores by class frequency, and weighted error F1 score (E. F1) does the same while restricting consideration to those examples with a runtime error.

We perform additional evaluations using the same experimental setup but distinct initializations to compute measures of variance, which we detail in Appendix F.

**RQ1:** The interpreter-inspired architectures show significant gains over the pylint, LSTM, GGNN and Transformer baseline approaches on the runtime error prediction task. We observe that the pylint baseline can make incorrect predictions because it correctly identifies an issue in the code under analysis when that code does not result in a runtime error in our dataset; pylint's lower performance on runtime error prediction is not evidence against pylint's performance for its intended use cases. We attribute the interpreter-inspired architectures' relative success over other neural architectures to their inductive bias toward mimicking program execution.

## 5.2 INCORPORATING RESOURCE DESCRIPTIONS

We next evaluate methods of incorporating resource descriptions into the models. For each architecture we apply the docstring approach of processing resource descriptions of Section 4.2. This completes a matrix of ablations, allowing us to distinguish the effects due to architecture change from the effect of the resource description. We follow the same experimental setup as in Section 5.1, and show the results again in Table 2a (compare rows with check marks to those without).

We also consider the FiLM and cross-attention methods of incorporating resource descriptions into the IPA-GNN. Following the same experimental setup again, we show the results of this experiment in Table 3. Note that the best model overall by our model selection criteria on validation data was the IPA-GNN with cross-attention, though the Exception IPA-GNN performed better on test.

**RQ2:** Across all architectures, the results show external resource descriptions improve performance on the runtime error prediction task. On the IPA-GNN architectures, we see further improvements

| MODEL | BASELINE | | | DOCSTRING | | | FILM | | | CROSS-ATTENTION | | |
|---|---|---|---|---|---|---|---|---|---|---|---|---|
| | ACC. | W. F1 | E. F1 | ACC. | W. F1 | E. F1 | ACC. | W. F1 | E. F1 | ACC. | W. F1 | E. F1 |
| IPA-GNN | 68.3 | 64.8 | 53.8 | 71.4 | 70.1 | 62.2 | 71.6 | 70.3 | 62.9 | 72.0 | 70.3 | 62.6 |
| E. IPA-GNN | 68.7 | 64.9 | 53.3 | 71.6 | 70.9 | 63.5 | 70.9 | 68.8 | 59.8 | 73.8 | 72.3 | 64.7 |

Table 3: A comparison of early and late fusion methods for incorporating external resource description information into interpreter-inspired models.

by considering architectures that incorporate the resource description directly into the execution step of the model, but these gains are inconsistent. The pylint baseline is unable to incorporate resource descriptions. Critically, using any resource description method is better than none at all.

To understand how the resource descriptions lead to better performance, we compare in Figure 2 the instruction pointer values of two Exception IPA-GNN models on a single example (shown in Table 4). The model with the resource description predicts that the `input()` calls will read input beyond the end of the stdin stream. In contrast, the model without the resource description has less reason to suspect an error would be raised by those calls. The descriptions of stdin in our runtime errors dataset also frequently reveal type information, expected ranges for numeric values, and formatting details about the inputs. We visualize additional examples in Appendix K.

## 5.3 INTERPRETABILITY AND LOCALIZATION

We next investigate the behavior of the Exception IPA-GNN model, evaluating its ability to localize runtime errors without any localization supervision. In unsupervised localization, the models predict the location of the error despite being trained only with error presence and kind supervision.

**Multiple Instance Learning Baselines** Unsupervised localization may be viewed as multiple instance learning (MIL) (Dietterich et al., 1997). Consider the subtask of predicting whether a particular line contains an error. In an $n$-line program, there are $n$ instances of this subtask. The available supervision only indicates if any one of these subtasks has an error, but not which one. By viewing each instance as a bag of subtasks, we have cast the problem as MIL.

Using this view, we introduce two variations on the Transformer architecture as multiple instance learning baselines. The first is the "Local MIL Transformer", in which each statement in the program is encoded individually, as in the local node embeddings computation of Section 4.1. The second is the "Global MIL Transformer", in which all tokens in the program may attend to all other tokens in the Transformer encoder. In both cases, the models make per-line predictions, which are aggregated to form an overall prediction as defined in Appendix I.

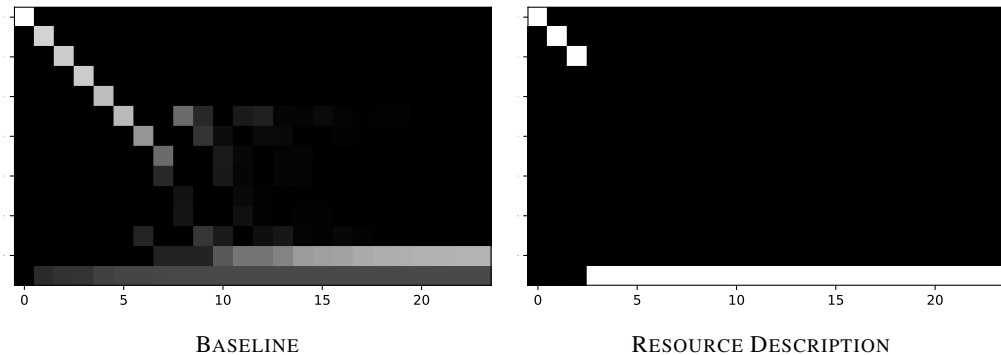

BASELINE                                      RESOURCE DESCRIPTION

Figure 2: Heatmap of instruction pointer values produced by BASELINE and DOCSTRING Exception IPA-GNNs for the example in Table 4. The x-axis represents timesteps and the y-axis represents nodes, with the last two rows respectively representing $n_{\text{exit}}$ and $n_{\text{error}}$. The BASELINE instruction pointer value is diffuse, with most probability mass ending at $n_{\text{exit}}$. The R.D. instruction pointer value is sharp, with almost all probability mass jumping to $n_{\text{error}}$ from node 2.

| | STDIN DESCRIPTION | Input: Input is given from Standard Input in the following format Constraints: Each character of S is A or B. \|S\| = 3 | | |
|---|---|---|---|---|
| $n$ | SOURCE | | BASELINE | R.D. |
| 0 | `a = str(input())` | | 16.9 | 0.4 |
| 1 | `q = int(input())` | | 3.2 | 0.3 |
| 2 | `s = [input().split() for i in range(q)]` | | 0.5 | **99.3** |
| 3,4 | `for i in range(q):` | | 6.4 | 0.0 |
| 5 | `    if int(s[i][0]) == 1 and len(a)>1:` | | 0.1 | 0.0 |
| 6 | `        a = a[::-1]` | | 0.7 | 0.0 |
| 7 | `    elif int(s[i][0])== 2 and int(s[i][1])==1:` | | 0.1 | 0.0 |
| 8 | `        a=s[i][2]+a` | | 0.2 | 0.0 |
| | `    else:` | | | |
| 9 | `        a=a+s[i][2]` | | 0.0 | 0.0 |
| 10 | `print(a)` | | 1.1 | 0.0 |

Table 4: Per-node localization predictions from the BASELINE and DOCSTRING Exception IPA-GNN models on a sample program from the validation split. The target class is EOFERROR, occurring on line 2 ($n = 2$). BASELINE predicts NO ERROR with confidence 0.708, while R.D. predicts EOFERROR with confidence 0.988, localized at line 3 ($n = 3$). The input description shows the cause for error: there are more `input()` calls than the number of expected inputs.

**Localization Experiment** Using the same protocol as Section 5.1, we train each of the MIL Transformer and Exception IPA-GNN models. As before, the models are trained only to minimize cross-entropy loss on predicting error kind and presence, receiving no error location supervision. We report the localization results in Table 2b. Localization accuracy ("LOCAL.") measures the percent of the test examples with errors for which the model correctly predicts the error line number.

**RQ3:** The Exception IPA-GNN's unsupervised localization capabilities far exceed that of baseline approaches. In Figure 2 we see the flow of instruction pointer mass during the execution of a sample program (Table 4) by two Exception IPA-GNN models, including the steps where the models raise probability mass to $n_{\text{error}}$. Tallying the contributions to $n_{\text{error}}$ from each node yields the exception provenance values in the right half of Table 4. This shows how the model's internal state resembles plausible program executions and allows for unsupervised localization. As a beneficial side-effect of learning plausible executions, the Exception IPA-GNN can localize the exceptions it predicts.

## 6 DISCUSSION

In this work, we introduce the task of predicting runtime errors in competitive programming problems and advance the capabilities of interpreter-inspired models. Our models support the complexity of competition code and demonstrate that natural language descriptions of external resources can reduce the ambiguity that arises in a static analysis setting. We show that the interpreter-inspired models outperform standard alternatives and that their inductive biases allow for interesting interpretability in the context of unsupervised localization.

Though they perform best, current IPA-GNN models require taking many steps of execution, up to 174 on this dataset. A future direction is to model multiple steps of program execution with a single model step, to reduce the number of model steps necessary for long programs. Extending the interpreter-inspired models with additional interpreter features, or supporting multi-file programs or programs with multiple user-defined functions are also interesting avenues for future work.

Learning to understand programs remains a rich area of inquiry for machine learning research because of its complexity and the many aspects of code. Learning to understand execution behavior is particularly challenging as programs grow in complexity, and as they depend on more external resources whose contents are not present in the code. Our work presents a challenging problem and advances interpreter-inspired models, both of which we hope are ingredients towards making progress on these difficult and important problems.

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

## A  Python Runtime Error Dataset Details

We describe in detail the construction of the Python Runtime Error dataset from the submissions in Project CodeNet (Puri et al., 2021). The Project CodeNet dataset contains over 14 million submissions to 4,053 distinct competitive programming problems, with the submissions spanning more than 50 programming languages. We partition the problems into train, valid, and test splits at an 80:10:10 ratio. By making all submissions to the same problem part of the same split we mitigate concerns about potential data leakage from similar submissions to the same problem. We restrict our consideration to Python submissions, which account for 3,286,314 of the overall Project CodeNet submissions, with 3,119 of the problems receiving at least one submission in Python. In preparing the dataset we execute approximately 3 million problems in a sandboxed environment to collect their runtime error information, we perform two stages of filtering on the dataset, syntactic and complexity filtering, and we construct a textual representation of the input space for each problem from the problem description.

### A.1  Syntactic Filtering

In this first phase of filtering, we remove submissions in Python 2 as well as those which fail to parse and run from our dataset. We remove 76,888 programs because they are in Python 2, 59,813 programs because they contain syntax errors that prohibit parsing, 2,011 programs that result in runtime errors during parsing, and 6 additional programs for which the python-graphs library fails to construct a control-flow graph. A program may result in a runtime error during parsing if it contains return, break, continue keywords outside of an appropriate frame.

### A.2  Program Execution

We attempt to run each submission in a sandboxed environment using the sample input provided in the Project CodeNet dataset. The environment is a custom harness running on a Google Cloud Platform (GCP) virtual environment. This allows us to collect standard out and standard error, to monitor for timeouts, and to catch and serialize any Python exceptions raised during execution. We restrict execution of each program to 1 second, marking any program exceeding this time as a timeout error. If the program encounters a Python exception, we use the name of that exception as the target class for the program. If an error type occurs only once in the dataset, we consider the target class to be Other. Programs not exhibiting an error or timeout are given target class "no error".

In addition to collecting the target class, we record for each runtime error the line number at which the error occurs. We use these line numbers as the ground truth for the unsupervised error localization task considered in Section 5.3.

### A.3  Extracting Resource Descriptions by Parsing Problem Statements

For each problem, we parse the problem statement to extract the *input description* and *input constraints*, if they exist. These two sections of the problem statement together form the external resource description that accompanies that problem. The problem statements in our dataset are each written either in English or Japanese, and so we write our parser to support both languages. When one or both of these sections are present in the problem statement, we construct the external resource description for the problem by concatenating together the headers and contents of the sections that are present. For the experiments that use the resource description as a docstring, we prepend to each submission a docstring containing the resource description for the problem that goes with that submission. Similarly these serve as the resource descriptions in the experiments that process resource descriptions via either cross-attention or FiLM.

### A.4  Vocabulary Construction and Complexity Filtering

All experiments use the same vocabulary and tokenization procedure. For this, we select the standard Byte-Pair Encoding (BPE) tokenization procedure (Sennrich et al., 2016). We construct the vocabulary using 1,000,000 submissions selected from the training split, along with the input space descriptions constructed for all problems in the train split. We use a vocabulary size of 30,000.

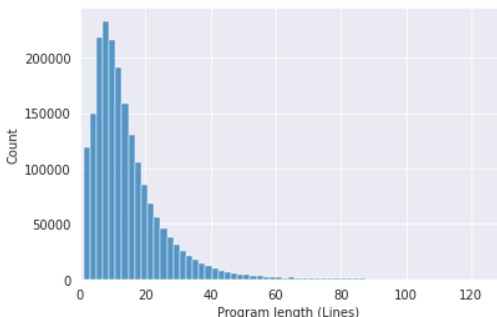 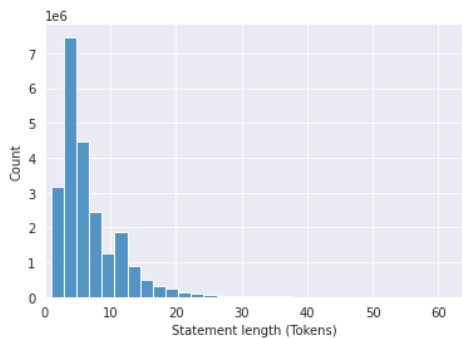

Figure 3: A histogram showing the distribution of program lengths, measured in lines, represented in the runtime errors dataset train split.

Figure 4: The distribution of statement lengths, measured in tokens, in the runtime errors dataset train split.

We then apply size-based filtering, further restricting the set of programs considered. First, the program length after tokenization is not to exceed 512 tokens, the number of nodes and edges in the control-flow graph are each not to exceed 128, and the step limit $T(x)$ for a program computed in Appendix E is not to exceed 174. We select these numbers to trim the long tail of exceptionally long programs, and this filtering reduces the total number of acceptable programs by less than 1%. To achieve consistent datasets comparable across all experiments, we use the longest form of each program (the program augmented with its input space information as a docstring) when computing the program sizes for size-based submission filtering.

We further impose the restriction that no user-defined functions (UDFs) are called in a submission; this further reduces the number of submissions by 682,220. A user-defined function is a function defined in the submission source code, as opposed to a built-in or a function imported from a third party module. Extending the IPA-GNN models to submissions with UDFs called *at most once* is trivially achieved by replacing the program's control-flow graph with its interprocedural control-flow graph (ICFG) (Nielson and Nielson, 1999). We leave the investigation of modeling user-defined functions to further work.

### A.5   FINAL DATASET DETAILS

After applying syntactic filtering (only keeping Python 3 programs that parse) and complexity filtering (eliminating long programs and programs that call user-defined functions), we are left with a dataset of 2,441,130 examples. The division of these examples by split and by target class is given in Table 1. Figure 3 shows the distribution of program lengths in lines represented in the completed dataset, with an average program length of 14.2 lines. The average statement length is 6.7 tokens, with the full distribution shown in Figure 4.

### A.6   DATA LICENSE

The Project CodeNet (Puri et al., 2021) data that we use is available under the CDLA Permissive v2.0 license, and we release our derived dataset under this same license.

### B   UNDER-APPROXIMATION OF ERROR LABELS

As described in Section 3, the ground truth error targets in our dataset are obtained by running each submission on only a single input. We do this because we only have a single input available from the online judges with which to execute the programs. As a result, the error labels we obtain under-approximate the full set of errors liable to appear at runtime. Metadata obtained from (Puri et al., 2021) indicates whether each submission encountered a runtime error on a larger set of inputs, though it does not indicate the kind or location of these errors when they are present. We use this metadata to determine the degree to which our labels are an under-approximation. We find that on the balanced test set there are 1,076 submissions (4%) which, per the metadata, encounter an error,

but for which our evaluation finds no error. These are likely examples for which the program runs without error on the input we have, but for which the program fails on some additional unavailable input.

We next measure generalization from the labels in our dataset to the labels suggested by the metadata without retraining. Since these labels are only binary indicators of error presence, we use our model to perform binary classification by summing the predicted probabilities of all error kinds. The model predicts "no error" on 76.2% of the examples for which our dataset finds no error. On the examples for which the metadata indicates no error, this drops to 75.9%, and on the examples for which the metadata indicates there is an error, this rises to 80.9%. These examples, where a single input detects no error but multiple inputs detect an error, are difficult for the model to classify. We hypothesize that the types of errors our labels omit systematically differ from those our labels include as an explanation for this 4.7% discrepancy.

## C    IPA-GNN ARCHITECTURE

We provide a concise and precise definition of the IPA-GNN baseline architecture, following the notation of Bieber et al. (2020). The IPA-GNN operates on the statement-level control-flow graph of the input program $x$, maintaining per-node per-step hidden states $h_{t,n}$ and a soft instruction pointer $p_{t,n}$. At each step $t$, each node $x_n$ participates in execution, branch prediction, and aggregation. First, the IPA-GNN models executing the statement at each node to produce per-node state proposals

$$a_{t,n}^{(1)} = \text{RNN}\left(h_{t-1,n}, \text{Embed}(x_n)\right). \tag{8}$$

Then, the model uses these to inform soft branch decisions at every control flow juncture, given as

$$b_{t,n,n_1}, b_{t,n,n_2} = \text{softmax}\left(\text{Dense}(a_{t,n}^{(1)})\right), \tag{9}$$

where $\{n_1, n_2\} = N_{\text{out}}(x_n)$ when $|N_{\text{out}}(x_n)| = 2$. When $|N_{\text{out}}(x_n)| = 1$ we have $b_{t,n,n'} = 1$ for $n' \in N_{\text{out}}(x_n)$ indicating straight-line code. For all other $n, n'$, $b_{t,n,n'} = 0$. The state proposals and branch decisions in turn feed into the computation of the new hidden states

$$h_{t,n} = \sum_{n' \in N_{\text{in}}(n)} p_{t-1,n'} \cdot b_{t,n',n} \cdot a_{t,n}^{(1)} \tag{10}$$

and new instruction pointer values

$$p_{t,n} = \sum_{n' \in N_{\text{in}}(n)} p_{t-1,n'} \cdot b_{t,n',n}. \tag{11}$$

The hidden state at final time step $T(x)$ at the program's exit node $n_{\text{exit}}$, given by $h_{T(x),n_{\text{exit}}}$ are used for downstream predictions.

## D    INPUT MODULATION

In Section 4.2, we consider both *cross-attention* (Lee et al., 2019) and *Feature-wise Linear Modulation* (FiLM) (Perez et al., 2017) as options for the $\text{Modulate}$ function. We provide the definitions of these operations here. First, cross-attention modules the input as:

$$\text{MultiHead}(\text{Embed}(x_n), d(x), h_{t-1,n}) = \text{Concat}(\text{Concat}(\text{head}_1, ..., \text{head}_h)W^O, \text{Embed}(x_n))$$
$$\tag{12}$$

$$\text{where head}_i = \text{softmax}\left(\frac{QK'}{\sqrt{d_k}}\right)V \tag{13}$$

$$Q = W_i^Q \text{Concat}(\text{Embed}(x_n), h_{t-1,n}) \tag{14}$$

$$K = W_i^K d(x) \tag{15}$$

$$V = W_i^V d(x) \tag{16}$$

Here, $W^O \in R^{hd_v \times d_{\text{model}}}$, $W_i^Q \in R^{d_k \times (d_{\text{model}} + d_{\text{Embed}(x_n)})}$, $W_i^K \in R^{d_k \times d_{d(x)}}$, and $W_i^V \in R^{d_v \times d_{d(x)}}$ are learnable parameters. Similarly, for FiLM we modulate the input with the resource description as follows:

$$\text{FiLM}(\text{Embed}(x_n), d(x), h_{t-1,n}) = \text{Concat}(\beta \cdot d(x) + \gamma, \text{Embed}(x_n)) \tag{17}$$
$$\text{where } \beta = \sigma(W_\beta \, \text{Concat}(x_n, h_{t-1,n}) + b_\beta), \tag{18}$$
$$\gamma = \sigma(W_\gamma \, \text{Concat}(x_n, h_{t-1,n}) + b_\gamma), \tag{19}$$

where $W_\gamma \in R^{d_{d(x)} \times (d_{\text{model}} + d_{\text{Embed}(x_n)})}$, and $W_\gamma \in R^{d_{d(x)} \times (d_{\text{model}} + d_{\text{Embed}(x_n)})}$ are learnable parameters.

# E TRAINING DETAILS

**Hyperparameter selection** We select hyperparameters by performing a random search independently for each model architecture. The hyperparameters considered by the search are listed in Table 6. All architectures use a Transformer encoder, and the Transformer sizes considered in the search are listed in Table 6 and defined further in Table 5.

| HYPERPARAMETER | T-128 | T-256 | T-512 |
|---|---|---|---|
| EMBEDDING DIMENSION | 128 | 256 | 512 |
| NUMBER OF HEADS | 4 | 4 | 8 |
| NUMBER OF LAYERS | 2 | 2 | 6 |
| QKV DIMENSION | 128 | 256 | 512 |
| MLP DIMENSION | 512 | 1024 | 2048 |

Table 5: Hyperparameter settings for each of the three Transformer sizes.

**Step limit** For the IPA-GNN and Exception IPA-GNN, the function $T(x)$ represents the number of execution steps modeled for program $x$. We reuse the definition of $T(x)$ from Bieber et al. (2020) as closely as possible, only modifying it to accept arbitrary Python programs, rather than being restricted to the subset of Python features considered in the dataset of the earlier work.

**Parameter counts** We provide in Table 7 the total number of parameters in each model, for the best performing hyperparameters in each model class. For all model classes, the maximum number of parameters considered is roughly equal (approximately 8.8 million).

**Compute usage and model speeds** All models are trained on Google Cloud Platform using TPUv2 accelerators. We use approximately one TPU-week of compute in training each IPA-GNN model. At inference time, IPA-GNN compute is proportional to the number of model steps, which is up to 174 for examples in our dataset. We measure the average inference time on the test set: 0.43

| HYPERPARAMETER | VALUE(S) CONSIDERED | ARCHITECTURE(S) |
|---|---|---|
| OPTIMIZER | {SGD} | ALL |
| BATCH SIZE | {32} | ALL |
| LEARNING RATE | {0.01, 0.03, 0.1, 0.3} | LSTM, TRANSFORMERS, IPA-GNNS |
| LEARNING RATE | {0.001, 0.003, 0.01, 0.03} | GGNN |
| GRADIENT CLIPPING | {0, 0.5, 1, 2} | ALL |
| HIDDEN SIZE | {64, 128, 256} | ALL |
| RNN LAYERS | {2} | LSTM, IPA-GNNS |
| GNN LAYERS | {8, 16, 24} | GGNN |
| SPAN ENCODER POOLING | {FIRST, MEAN, MAX, SUM} | ALL |
| CROSS-ATTENTION NUMBER OF HEADS | {1, 2} | IPA-GNNS WITH CROSS-ATTENTION |
| MIL POOLING | {MAX, MEAN, LOGSUMEXP} | MIL TRANSFORMERS |
| TRANSFORMER DROPOUT RATE | {0, 0.1} | ALL |
| TRANSFORMER ATTENTION DROPOUT RATE | {0, 0.1} | ALL |
| TRANSFORMER SIZE | {T-128, T-256, T-512} | ALL |

Table 6: Hyperparameters considered for random search during model selection.

| MODEL | PARAMETER COUNT | TRAIN LATENCY | INFERENCE LATENCY |
|---|---|---|---|
| GGNN | 4,831,903 | 0.055 | 0.040 |
| TRANSFORMER | 8,578,975 | 0.054 | 0.051 |
| LSTM | 4,361,823 | 0.058 | 0.057 |
| IPA-GNN | 4,368,161 | 0.727 | 0.294 |
| E. IPA-GNN | 8,856,099 | 1.167 | 0.435 |

Table 7: The parameter count, training latency (sec/step), and inference latency (sec/batch) for the best performing instance of each model variant. Training and inference latencies use batch size 32.

seconds per batch of 32. We also measure the training speed in seconds per step for each method, which we report in Table 7. We observe that the IPA-GNN train times are slower than those of the generic models, a drawback of the IPA-GNN model family in its current implementations. That said, we also note that the IPA-GNN models do not benefit from the same optimizations as basic implementations of the well known general purpose models (GGNN, Transformer, and LSTM), and with further optimizations the IPA-GNN performance can be improved.

## F   METRIC VARIANCES

Under the experimental conditions of Section 5.1, we perform three additional training runs to calculate the variance for each metric for each baseline model, and for the Exception IPA-GNN model using the docstring strategy for processing resource descriptions. For these new training runs, we use the hyperparameters obtained from model selection. We vary the random seed between runs (0, 1, 2), thereby changing the initialization and dropout behavior of each model across runs. We report the results in Table 8; $\pm$ values are one standard deviation.

| METHOD | R.D.? | ACC. | W. F1 | E. F1 |
|---|---|---|---|---|
| GGNN | | $61.98 \pm 1.24$ | $56.62 \pm 2.96$ | $41.24 \pm 5.51$ |
| TRANSFORMER | | $63.82 \pm 0.62$ | $59.86 \pm 0.52$ | $46.75 \pm 0.93$ |
| LSTM | | $66.43 \pm 0.60$ | $62.33 \pm 1.12$ | $50.10 \pm 1.94$ |
| EXCEPTION IPA-GNN | ✔ | $71.44 \pm 0.15$ | $70.78 \pm 0.07$ | $63.54 \pm 0.03$ |

Table 8: Mean and standard deviation for each metric is calculated from three training runs per model, using the hyperparameters selected via model selection.

## G   STATIC ANALYSIS BASELINE

Our work builds towards a developer tool that predicts runtime errors in programs without running the program, treating the task as static analysis. Existing static analysis tools already inspect Python source code for possible issues, though they are not generally designed with runtime error prediction in mind. Among the most popular such tools are the linters pylint and flake8, the type analyzer pytype, and the formatter black. We elect to compare against pylint as it is the most common of these tools and hence most representative of a modern developer workflow. Additionally, a formatter is not well suited to the task of predicting errors, and type analysis benefits from type annotations which are rarely utilized in competition code. In our comparison of machine learning methods against pylint (Section 5), we build a runtime error classifier based on pylint's output. For each kind of error or warning that pylint can detect, we determine whether it is indicative a runtime error class. For example, pylint's error `no-member (E1101)` indicates the AttributeError runtime error. The pylint baseline predicts a runtime error class whenever pylint's errors or warnings indicate that error class, and "no error" otherwise. Table 9 shows the mapping from pylint findings to runtime error.

Only eleven of the twenty-six runtime error classes (those listed in Table 9, and "no error") can be predicted by this baseline. Additionally, the presence of a pylint finding that corresponds to an error does not guarantee the error will actually be present when running the program; for example

| Error Class | PyLint Finding |
|---|---|
| AssertionError | bad-thread-instantiation (W1506) |
| AttributeError | misplaced-format-function (E0119) |
| | no-member (E1101) |
| | not-context-manager (E1129) |
| | missing-format-attribute (W1306) |
| | not-async-context-manager (E1701) |
| ImportError | import-error (E0401) |
| | relative-beyond-top-level (E0402) |
| | no-name-in-module (E0611) |
| IndexError | potential-index-error (E0643) |
| | too-few-format-args (E1306) |
| | invalid-format-index (W1307) |
| KeyError | missing-format-argument-key (W1303) |
| | missing-format-string-key (E1304) |
| NameError | used-before-assignment (E0601) |
| | undefined-variable (E0602) |
| RuntimeError | misplaced-bare-raise (E0704) |
| | modified-iterating-dict (E4702) |
| | modified-iterating-set (E4703) |
| SyntaxError | syntax-error (E0001) |
| | return-outside-function (E0104) |
| | yield-outside-function (E0105) |
| | duplicate-argument-name (E0108) |
| | too-many-star-expressions (E0112) |
| | invalid-star-assignment-target (E0113) |
| | star-needs-assignment-target (E0114) |
| | nonlocal-and-global (E0115) |
| | nonlocal-without-binding (E0117) |
| | used-prior-global-declaration (E0118) |
| | await-outside-async (E1142) |
| | yield-inside-async-function (E1700) |
| | invalid-unicode-codec (E2501) |
| | bidirectional-unicode (E2502) |
| TypeError | abstract-class-instantiated (E0110) |
| | bad-reversed-sequence (E0111) |
| | invalid-slots-object (E0236) |
| | invalid-slots (E0238) |
| | inherit-non-class (E0239) |
| | inconsistent-mro (E0240) |
| | duplicate-bases (E0241) |
| | invalid-enum-extension (E0244) |
| | invalid-length-returned (E0303) |
| | invalid-bool-returned (E0304) |

| Error Class | PyLint Finding |
|---|---|
| TypeError (cont.) | invalid-index-returned (E0305) |
| | invalid-repr-returned (E0306) |
| | invalid-str-returned (E0307) |
| | invalid-bytes-returned (E0308) |
| | invalid-hash-returned (E0309) |
| | invalid-length-hint-returned (E0310) |
| | invalid-format-returned (E0311) |
| | invalid-getnewargs-returned (E0312) |
| | invalid-getnewargs-ex-returned (E0313) |
| | unpacking-non-sequence (E0633) |
| | raising-bad-type (E0702) |
| | bad-exception-cause (E0705) |
| | raising-non-exception (E0710) |
| | notimplemented-raised (E0711) |
| | catching-non-exception (E0712) |
| | bad-super-call (E1003) |
| | not-callable (E1102) |
| | isinstance-...-not-valid-type (W1116) |
| | no-value-for-parameter (E1120) |
| | too-many-function-args (E1121) |
| | unexpected-keyword-arg (E1123) |
| | redundant-keyword-arg (E1124) |
| | missing-kwoa (E1125) |
| | invalid-sequence-index (E1126) |
| | invalid-slice-index (E1127) |
| | invalid-unary-operand-type (E1130) |
| | unsupported-binary-operation (E1131) |
| | repeated-keyword (E1132) |
| | not-an-iterable (E1133) |
| | unsupported-membership-test (E1135) |
| | unsubscriptable-object (E1136) |
| | unsupported-assignment-operation (E1137) |
| | unsupported-delete-operation (E1138) |
| | dict-iter-missing-items (E1141) |
| | unhashable-member (E1143) |
| | bad-format-character (E1300) |
| | mixed-format-string (E1302) |
| | format-needs-mapping (E1303) |
| | bad-string-format-type (E1307) |
| | invalid-envvar-value (E1507) |
| | invalid-envvar-default (W1508) |
| ValueError | return-in-init (E0101) |
| | class-variable-slots-conflict (E0242) |
| | unbalanced-tuple-unpacking (W0632) |
| | bad-format-string (W1302) |
| | format-combined-specification (W1305) |
| | bad-open-mode (W1501) |

Table 9: The pylint baseline for runtime error prediction predicts the error class shown when it encounters any of the corresponding pylint findings. Many of pylint's 235 finding types do not indicate runtime errors. This table shows the mapping used by the pylint baseline.

an undefined variable may appear on an unused control-flow path, benign at runtime. The results of this baseline are reported in Section 5.

# H    Localization by Modeling Exception Handling

For programs that lack try/except frames, we compute the localization predictions of the Exception IPA-GNN model by summing, separately for each node, the contributions from that node to the global error node across all time steps. This gives an estimate of exception provenance as

$$p(\text{error at statement } n) = \sum_t p_{t,n} \cdot b_{t,n,n_{\text{error}}}. \tag{20}$$

For programs with a try/except frame, however, we must trace the exception back to the statement that originally raised it. To do this, we keep track of the exception provenance at each node at each time step; when an exception raises, it becomes the exception provenance at the statement that it raises to, and when a statement with non-zero exception provenance executes without raising, it propagates its exception provenance information to the next node unchanged.

Define $v_{t,n,n'}$ as the amount of "exception probability mass" at time step $t$ at node $n'$ attributable to an exception starting at node $n$. Then we write

$$v_{t,n,n'} = \sum_{k \in N_{\mathrm{in}}(n')} v_{t-1,n,k} \cdot b_{t,k,n'} \cdot p_{t,k} + \left(1 - \sum v_{t-1,:,n}\right) \cdot b_{t,n,n'} \cdot p_{t,n} \cdot \mathbb{1}\{n' = r(n)\}. \quad (21)$$

The first term propagates exception provenance across normal non-raising execution, while the second term introduces exception provenance when an exception is raised. We then write precisely

$$p(\text{error at statement } n) = v_{T(x),n,n_{\mathrm{error}}}, \quad (22)$$

allowing the Exception IPA-GNN to make localization predictions for any program in the dataset.

## I   LOCALIZATION BY MULTIPLE INSTANCE LEARNING

The Local Transformer and Global Transformer models each compute per-statement node embeddings $\mathrm{Embed}(x_n)$ given by Equation 1. In the multiple instance learning setting, these are transformed into unnormalized per-statement class predictions

$$\phi(\text{class} = k, \text{lineno} = l) = \mathrm{Dense}(\mathrm{Embed}(x_n)). \quad (23)$$

We consider three strategies for aggregating these per-statement predictions into an overall prediction for the task. Under the *logsumexp* strategy, we treat $\phi$ as logits and write

$$\log p(\text{class} = k) \propto \log\left(\sum_l \exp \phi(k,l)\right), \quad (24)$$

$$\log p(\text{lineno} = l) \propto \log\left(\sum_{k \in K} \exp \phi(k,l)\right) \quad (25)$$

where K is the set of error classes.

The *max* and *mean* strategies meanwhile follow Wang et al. (2018) in asserting

$$p(\text{class} = k \mid \text{lineno} = l) = \mathrm{softmax}(\phi(k,l)), \quad (26)$$

compute the location probabilities as

$$p(\text{lineno} = l) \propto \sum_{k \in K} p(\text{class} = k \mid \text{lineno} = l), \quad (27)$$

and compute the outputs as

$$\log p(\text{class} = k) \propto \log \max_l p(\text{class} = k \mid \text{lineno} = l), \text{ and} \quad (28)$$

$$\log p(\text{class} = k) \propto \log \frac{1}{L} \sum_l p(\text{class} = k \mid \text{lineno} = l) \quad (29)$$

respectively, where $L$ denotes the number of statements in $x$. As with all methods considered, the MIL models are trained to minimize the cross-entropy loss in target class prediction, but these methods still allow reading off predictions of $p(\text{lineno})$.

## J   BROADER IMPACT

Our work builds toward improvements to developer tools, suggesting the possibility of future tools that predict runtime errors in code even when that code lacks unit tests. However, the false positive rate under the current best models present a challenge. A developer tool built using these models may present the developer with incorrect predictions. This could cause the developer to make mistakes, or to lose trust in the tooling, lowering productivity in the short term and making it harder to win back trust in the long term when tools are built upon higher quality models with fewer errors. We therefore recommend that tool developers use a combination of cautious judgement and data driven evaluations when deciding when to implement features that rely on models like the ones we present.

## K    EXAMPLE VISUALIZATIONS

We sample three examples at random from the Python Runtime Error dataset validation split, and visualize them here. As in Figure 2, we show instruction pointer heatmaps for both the BASELINE and DOCSTRING Exception IPA-GNN model variants. The values in the tables show the total error contribution predicted by the model (Appendix H) associated with each line of code.

In the heatmaps, the x-axis represents time steps and the y-axis represents nodes, with the last two rows representing the exit node $n_{\text{exit}}$ and the error node $n_{\text{error}}$. Note that for loop statements are associated with two spans in the statement-level control-flow graph, one for construction of the loop iterator, and a second for assignment to the loop variable. Hence we list two indexes for each for loop statement in these figures, and report the total error contribution for the line.

| STDIN DESCRIPTION | Input:  Input is given from Standard Input in the following format:  N a_1 a_2 ...  a_N Constraints:  All values in input are integers.  1 <= N , a_i <= 100 |
|---|---|

| $n$ | SOURCE | BASELINE | R.D. |
|---|---|---|---|
| 0 | `N = int(input())` | 2.9 | 0.2 |
| 1 | `A = list(map(int, input().split()))` | 0.8 | 0.0 |
| 2 | `res = 0` | 3.0 | **63.3** |
| 3,4 | `for i in range(1, len(A)+1, 2):` | 9.8 | 6.3 |
| 5 | `    res += A[i] % 2` | 0.3 | 0.1 |
| 6 | `print(res)` | 0.2 | 2.2 |

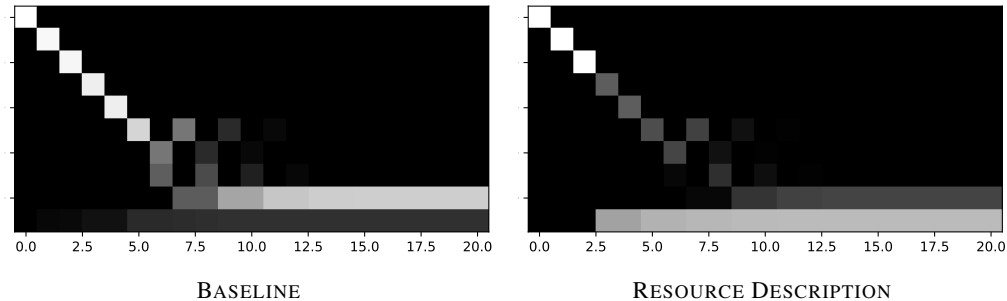

BASELINE                                    RESOURCE DESCRIPTION

Figure 5: The target error kind is INDEXERROR, occuring on line 5 ($n = 5$). BASELINE incorrectly predicts NO ERROR with confidence 0.808. DOCSTRING correctly predicts INDEXERROR with confidence 0.693, but localizes to line 3 ($n = 2$). Both BASELINE and DOCSTRING instruction pointer values start out sharp and become diffuse when reaching the for-loop. The BASELINE instruction pointer value ends with most probability mass at $n_{\text{exit}}$. The DOCSTRING instruction pointer value has a small amount of probability mass reaching $n_{\text{exit}}$, with most probability mass ending at $n_{\text{error}}$.

| | STDIN DESCRIPTION | Input: Input is given from Standard Input in the following format: H N A_1 A_2 ... A_N Constraints: 1 <= H <= 10^9 1 <= N <= 10^5 1 <= A_i <= 10^4 All values in input are integers. | | |
|---|---|---|---|---|
| $n$ | SOURCE | | BASELINE | R.D. |
| 0 | `H,N,A = list(map(int, input().split()))` | | 9.7 | 3.4 |
| 1,2 | `for i in A[N]:` | | **43.7** | **83.0** |
| 3 | `  if H <= 0:` | | 2.9 | 2.8 |
| | `    break` | | | |
| | `  else:` | | | |
| 4 | `    H -= A[i]` | | 6.0 | 0.0 |
| 5 | `if set(A):` | | 0.2 | 0.1 |
| 6 | `  print("Yes")` | | 9.3 | 0.7 |
| | `else:` | | | |
| 7 | `  print("No")` | | 3.3 | 0.2 |

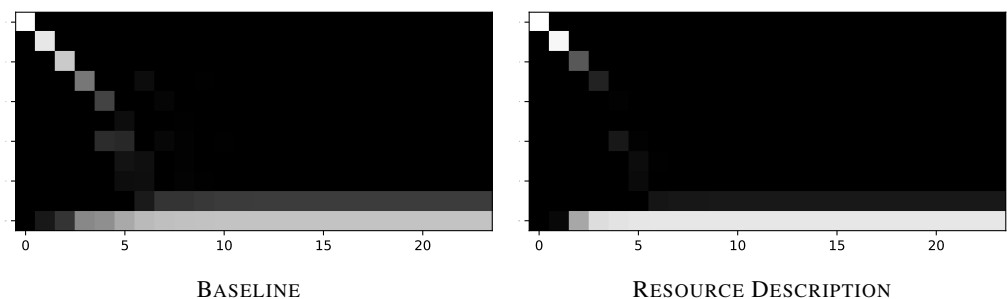

BASELINE                    RESOURCE DESCRIPTION

Figure 6: The target error kind is VALUEERROR, occuring on line 1 ($n = 0$). BASELINE incorrectly predicts INDEXERROR with confidence 0.319 on line 1 ($n = 0$). DOCSTRING correctly predicts VALUEERROR with confidence 0.880 on line 2 ($n = 1$), corresponding to `A[n]`. Both BASELINE and DOCSTRING instruction pointer values start out sharp and quickly shift most of the probability mass to the error node.

| STDIN DESCRIPTION | Input: n m d1 d2 ... dm Two integers n and m are given in the first line. The available denominations are given in the second line. Constraints: 1 <= n <= 50000 1 <= m <= 20 1 <= denomination <= 10000 The denominations are all different and contain 1. |
|---|---|

| $n$ | SOURCE | BASELINE | R.D. |
|---|---|---|---|
| 0 | `from itertools import combinations_with_replacement as C` | 40.1 | 1.3 |
| 1 | `n, m = map(int, input().split())` | 2.3 | 7.1 |
| 2 | `coin = sorted(list(map(int, input().split())))` | 7.2 | 2.8 |
| 3 | `if n in coin:` | 2.0 | 0.2 |
| 4 | `  print(1)` | 2.0 | 1.5 |
|   | `else:` | | |
| 5 | `  end = n // coin[0] + 1` | 0.3 | 0.1 |
| 6 | `  b = False` | 0.1 | 0.3 |
| 7,8 | `  for i in range(2, end):` | 2.4 | 0.7 |
| 9,10 | `    for tup in list(C(coin, i)):` | 3.4 | 1.2 |
| 11 | `      if sum(tup) == n:` | 0.3 | 0.0 |
| 12 | `        print(i)` | 0.3 | 0.1 |
| 13 | `        b = True` | 0.6 | 0.9 |
|   | `        break` | | |
| 14 | `  if b: break` | 0.1 | 1.4 |

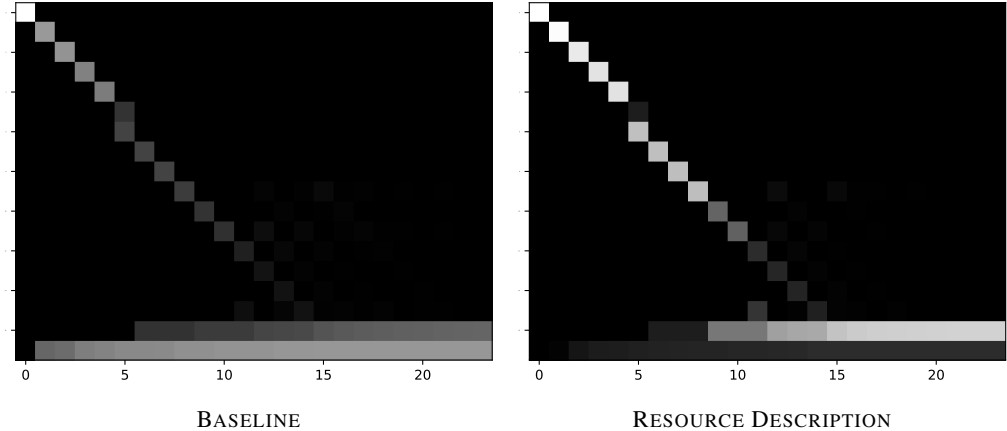

BASELINE                    RESOURCE DESCRIPTION

Figure 7: The target error kind is NO ERROR. BASELINE correctly predicts NO ERROR with confidence 0.416. DOCSTRING also correctly predicts NO ERROR with confidence 0.823. The BASELINE instruction pointer value makes its largest probability mass contribution to $n_{\text{error}}$ at $n = 0$ and ends up with mass split between $n_{\text{exit}}$ and $n_{\text{error}}$. The DOCSTRING instruction pointer value accumulates little probability in $n_{\text{error}}$ and ends up with most probability mass in $n_{\text{exit}}$.

