# OpenReview forum: "Static Prediction of Runtime Errors by Learning to Execute Programs with External Resource Descriptions"
_ICLR.cc/2023/Conference — ICLR 2023 poster_

### Official Review · Reviewer_Gdcd · 2022-10-24

**Confidence:** 3
**Correctness:** 3
**Technical Novelty And Significance:** 3
**Empirical Novelty And Significance:** 2
**Recommendation:** 6

**Clarity, Quality, Novelty And Reproducibility:**

The paper is well written.
I think that the IPA-GNN extensions and improvements are novel.
The paper results should be reproducible using the dataset authors plan to release and the descriptions of the code/methods in the paper. The reproducibility might improve if the code was also released by authors.

-------------------------------------------------------------------------------------------------------
ICLR review form does not provide a section for comments/questions about the paper. I am going to post them here:
- Authors should clarify that the "external resource descriptions" could be comments, problem statement, possible inputs, expected outputs and so on. They also perhaps should analyze and comment on which of these descriptions work better or at all.

**Strength And Weaknesses:**

Strengths:

- Improvement of IPA-GNN model to handle exceptions and real programs
- Adding handling of resource descriptions to IPA-GNN
- Unsupervised localization of errors
- Ablation analysis
- Improved results compared to baselines and ablations

Weaknesses:

- Authors compare results to baselines, but do not try to identify state of the art results for their testset and compare against state of the art methods


**Summary Of The Paper:**

Authors propose an approach to predict runtime errors of programs by using IPA-GNN models with improvements to handle exceptions and provide error locations. They also add support for passing resource descriptions to the model. Authors provide a dataset for testing their approach. They show that their approach outperforms baselines.

**Summary Of The Review:**

I think that authors present a novel improvement of IPA-GNN to handle exceptions, real programs and external descriptions. Their approach improves prediction of runtime errors vs baselines and I think this may warrant accept to ICLR.

---

> ### Author Response · Authors · 2022-11-18
> **Response to Reviewer Gdcd**
>
> Thanks for your review! We appreciate your concise yet thorough summary of the main strengths / contributions of our paper. Let us respond to the points that you raised.
>
> > Authors compare results to baselines, but do not try to identify state of the art results for their testset and compare against state of the art methods
>
> Thanks for identifying this point about our paper. Since this is a new task focused on a new class of errors, there is not a clearly applicable state of the art method from the literature. For example, DeepBugs [1] is a strong approach in a related bug prediction task, but it considers only a single line at a time. (We note that this could be extended to predicting runtime errors in a full submission via multiple instance learning resulting in an approach similar to the Local MIL Transformer that we consider in 5.3.)
>
> We selected several well recognized baselines to compare against (Transformer, LSTM, GGNN), including Transformer which achieves state of the art performance on a wide range of tasks in many domains. Then, rather than identifying the best possible performing model (which would likely require larger model scales and additional compute), we are careful to hold the model scales (number of parameters) approximately fixed across different model types in our investigations (Appendix E). We also evaluated pylint as a baseline in order to provide a comparison with the standard Python toolchain.
>
> > Authors should clarify that the "external resource descriptions" could be comments, problem statement, possible inputs, expected outputs and so on. They also perhaps should analyze and comment on which of these descriptions work better or at all.
>
> In Section 3 we list inputs, file contents, and network access as possible external resources. However, for our competitive programming dataset, the only external resource the programs depend on is stdin. Therefore, descriptions of stdin are the only external resource we have evaluated so far. We take your point that it will be valuable to evaluate these alternatives in future work.
>
> > The reproducibility might improve if the code was also released by authors.
>
> We commit to releasing the code in addition to the dataset.
>
> [1] Michael Pradel, Koushik Sen. DeepBugs: A Learning Approach to Name-based Bug Detection

---

### Official Review · Reviewer_NFFC · 2022-10-27

**Confidence:** 3
**Correctness:** 4
**Technical Novelty And Significance:** 4
**Empirical Novelty And Significance:** 4
**Recommendation:** 8

**Clarity, Quality, Novelty And Reproducibility:**

**Clarity :**

This paper attempts to convey a very sophisticated method with several "moving parts". Considering that, it is written extremely clearly and concisely. But I still have some questions about certain aspects of the algorithm. I've asked questions to clarify these under the headline of "Suggestions/Clarifications" above.

**Quality and Novelty :**

The experiments are thorough (with several auxiliary details in the Appendix), the design choices are well-motivated, the idea is novel, and the results are very impressive, especially on "unsupervised" error localization.

**Reproducibility :**

The authors have mentioned that their models and dataset will be made available after the review process, so it is not possible to check their reproducibility claims at this time. But the description of experiments in the paper are thorough enough that it should be possible to re-implement their approach with some (possibly significant) engineering effort.

**Strength And Weaknesses:**

**Strengths :**

1. The extensions from the previous IPA-GNN paper are novel and non-trivial.

1. The evaluation is thorough and the results are convincing. The results on error localization in particular are very impressive considering that the model was not trained explicitly for that task.

1. Extensive details have been provided in the appendix about the experimental setup. Further, I appreciate the principled experiment design with carefully constructed metrics (weighted F1, weighted error F1, etc), and model selection performed on validation data as it should be.

**Weaknesses :**

1. I'm unsure if pylint is an appropriate choice for a static analysis baseline. Wouldn't something based on symbolic execution be closer to the spirit of this paper? After all, this model attempts to simulate the execution of the program, which is closer to symbolic execution than static analysis, in my opinion. Further, it is only natural that static analysis cannot catch runtime errors effectively, whereas symbolic execution approaches might be able to find values that throw exceptions.

1. How does this scale with the size of the program? The programs in CodeNet are still relatively small programs compared to kernel code etc which could be hundreds of thousands of lines. What is the execution time/memory for the largest programs in CodeNet? And how does the graph of computation cost with size of program look?

**Suggestions/Clarifications :**
(I feel like it wouldn't be accurate to call these points "Weaknesses")

*Point 1*

Regarding Table 4, here is my attempt to understand what's going on : `|S| = 3`, so if `q` is some number `>3`, we could have an EOF error when it tries to read from stdin more than 3 times. The model sees the constraint `|S| = 3` as part of the stdin description, and that is why it is able to predict this accurately.

Could something like the above ^ be put in the paper, either in the table caption or in the text? I know that the end of Section 5.2 has a brief note to this effect, but it would really be nice if the part about `|S| = 3` could be emphasized as that is the key bit of info that helps the model make the decision (at least as per my understanding).

*Point 2*

Regarding the example in Table 4, I feel it would be interesting to evaluate the model with the same resource description, but with `|S| = 3` removed. It would show that *that* is the piece of information that the model is using to make a better prediction.

*Point 3*

When you use a Transformer to generate embeddings of each token (and to embed the resource description) as the first step of E IPA-GNN, do you use a pre-trained Transformer model? What architecture? Do you freeze the weights or do you learn it along with the rest of the model? It would be useful if this information was provided.

*Point 4*

I find the flow of RQ1 and RQ2 a little confusing in the context of Table 2a. When we first read RQ1, it's not apparent that the method of incorporating resource descriptions is as a docstring. Then later in RQ2 that is specified, so we have to jump back to Table 2a to consult it again. I feel like there should be a note in RQ1 specifying that docstrings are used, and other methods like FiLM and Cross-attention will be discussed in the next RQ.

*Point 5*

Does IPA-GNN (not E.IPA-GNN) also have an $n_{\text{error}}$ state?

If not, how do you use IPA-GNN to predict errors? For E. IPA-GNN the method is clear - use a dense layer on the $n_{\text{error}}$ state in the last timestep. For IPA-GNN, what state do you use?

Similarly for localization. How do you localize a bug using IPA-GNN if there is no $n_{\text{error}}$ state?

**Summary Of The Paper:**

This paper tackles the problem of identifying runtime errors in a program in a static setting. Their proposed model is a modification of an Instruction Pointer Attention GNN (IPA-GNN) proposed in Bieber et al. (2020) [1], which learns to execute the program one instruction at a time in a "continuous" manner using embeddings and probabilistic transitions. They train this model on the problem of classifying a program into either a) one of many defined error classes, or b) no error. Then they show that they are able to use the inner states of the model to localize runtime errors occurring in a specific statement, with impressive accuracy.

Since the IPA-GNN [1] model forms the basis for this work, I provide a quick summary of my understanding of this paper.

------------------
**<IPA-GNN>**

Each statement in a (simple) program is represented with a 4-tuple that constitutes the statement's initial embedding. The execution is simulated in a series of time steps (like an RNN). The model maintains a probabilistic instruction pointer, where $p_{t, n}$ represents the "weight" given to statement $n$ at time step $t$. Similarly, there is a hidden state embedding $h_{t, n}$ for each time step and statement. For a time step $t$ and statement $n$, the "state proposal" is computed as
$$ a_{t, n} = \text{RNN} \left( h_{t, n}, \text{Embed}(x_n) \right) $$

Then :

* To compute the next time step, the model predicts a branch probability using a Dense layer on the state proposal. This is unclear in the paper, but I assume there can only be at most 2 possible next states. I also assume that the Dense layer weights are shared among all such branch predictions.

* The next hidden states $h_{t+1, n}$ and instruction pointer probabilities $p_{t+1, n}$ are computed using a weighted sum over all branch paths that lead *into* $n$, weighted by both the instruction pointer probabilities and the branch probabilities.

**End of </IPA-GNN>**

------------------

The authors of this paper build on the IPA-GNN in the following ways :

1. They perform their analyses on a large annotated dataset of real Python programs, possibly with runtime errors.

1. Instead of a 4 tuple to represent a statement, they run a Transformer over the token embeddings and pool to get an embedding for each statement.

1. Since these programs are long, the computational graph could be very large. So they apply a "rematerialization" trick at each layer to make the model memory efficient (I did not make an effort to understand the details of this).

1. They take a natural language description of the stdin input format and construct an embedding from it using the same Transformer from earlier. This is then combined ("modulated") with the statement embedding at each statement and each time step.

1. Before predicting a branch, they use a separate Dense layer to predict whether there is an exception at each statement. All thrown exceptions lead to a special $n_{\text{error}}$ node.
------------------

The authors then evaluate their approach on a) classifying a program into one of various error types (or no error), and b) localizing the statement corresponding to a runtime exception. They show significant gains over their baselines on these tasks. Further, they perform an interpretability study on one handpicked example program.

[1] David Bieber, Charles Sutton, Hugo Larochelle, and Daniel Tarlow. Learning to execute program with instruction pointer attention graph neural networks. In Advances in Neural Information Processing Systems, 2020.

**Summary Of The Review:**

It is my opinion that this paper presents a novel idea convincingly and elaborately. I have a few clarifying questions about the details of their algorithm/experiments, along with a minor concern about the choice of one of their baselines. But on the whole, I feel that the paper, even in its current form, is an excellent contribution to the program analysis literature and therefore I wholeheartedly recommend acceptance.

I give myself a confidence score of 3/5 **not** because I don't understand the paper well-enough, but because I am not sufficiently familiar with the current research in this area to know if there are other potential competing approaches to solve this problem.

---

> ### Author Response · Authors · 2022-11-18
> **Response to Reviewer NFFC**
>
> Thank you for your review. We agree with your summary of the paper, and appreciate that you have taken the time to summarize [1] as well. The assumptions that you make in summarizing [1] are correct. We have verified these with the [original codebase on GitHub](https://github.com/google-research/google-research/blob/911a27b5377de365887d6e73dc470ade5dfcf527/ipagnn/models/ipagnn.py#L110-L114).
> We will now respond to the two weaknesses you identified and the five suggestions and clarifications.
>
> > I'm unsure if pylint is an appropriate choice for a static analysis baseline.
>
> Our selection of pylint as a static analysis baseline is practically motivated. It is a standard tool in a Python development toolchain. That said, we agree symbolic execution is interesting here. However, symbolic execution approaches often require nontrivial program modifications and do not support the full range of programs considered in the competitive programming dataset.
>
> > How does this scale with the size of the program?
>
> The largest programs in the dataset that we consider are 512 tokens after tokenization. There is a long tail of longer programs that we have excluded from Project CodeNet in preparing this dataset, but removing these longer programs reduces the dataset size by less than 1% (see Appendix A.4). We follow [1] to determine T(x) the number of steps of IPA-GNN that the model runs for. The runtime of the model is proportional to T(x) times the number of nodes in the program's control flow graph. The average inference time is 0.435 seconds for a batch of 32 programs for our best model. We have plotted the number of model steps T(x) as a function of program length and added it to Appendix E as Figure 5. T(x) does grow as program length increases, though it is more closely related to program complexity than program length; the largest value of T(x) among any of the programs considered is 174. Scaling to much larger programs, like kernel code, is going to require new approaches beyond the scope of this paper.
>
> Point 1 and Point 2 (Table 4):
> >  |S| = 3, so if q is some number >3...
>
> It is actually more direct than that. The input to this program is a single line consisting of |S| = 3 characters (to your point 2: the "|S| = 3" part of the description is not critical to determining the failure). The program incorrectly expects there to be multiple lines of input, including a number q (but the input doesn't actually contain a second line of input with a number q.) The resource description model correctly predicts that the second call to input() yields an EOFError since the input is just one line. In investigating this, we found a bug in our visualization code that produced Table 4 and the tables in Appendix K, and we have updated them accordingly. We have also updated the Table 4 caption to clarify the behavior here.
>
> Point 3:
>
> The transformer is initialized from scratch. We use the flax transformer from https://github.com/google/flax/tree/main/examples. The weights are learned along with the rest of the model.
>
> Point 4:
>
> Thank you for the clarity suggestions. In RQ1, we are not yet considering resource descriptions. Nevertheless, Table 2 does include results with resource descriptions, though those results are not referenced until RQ2. We have added an explanatory note to improve the flow and clarity of this.
>
> Point 5:
>
> IPA-GNN does not have an $n_\text{error}$ state. Following [1], the IPA-GNN predicts errors using a dense layer on the final state (time step T) at $n_\text{exit}$. This does not readily admit localization, and accordingly we do not include the vanilla IPA-GNN in our localization experiment (Table 2b).
>
> Localization with IPA-GNN could be obtained using a strategy similar to multiple instance learning. We would apply the IPA-GNN giving every node a corresponding exit node that accumulates the probability mass and hidden states passing through its corresponding interior node. Predicting the error at each exit node and aggregating as with multiple instance learning (Section 5.3) would allow localization predictions using the IPA-GNN. This approach is interesting to consider but we have not experimented with it.

---

> > ### Comment · Reviewer_NFFC · 2022-12-06
> > **Response to the authors**
> >
> > Thank you for your responses and for satisfactorily addressing my comments.

---

### Official Review · Reviewer_xTjd · 2022-11-03

**Confidence:** 4
**Correctness:** 2
**Technical Novelty And Significance:** 3
**Empirical Novelty And Significance:** 3
**Recommendation:** 6

**Clarity, Quality, Novelty And Reproducibility:**

The paper is clearly written and easy to follow. The paper has moderate novelty due to the reasons mentioned above.

**Strength And Weaknesses:**

Strengths:
1.	This paper an interesting problem of locating runtime error with neural interpreter models. The proposed dataset may be useful for the future study in the area.
2.	The proposed approach can be used for locating the bug, even though the model is trained with only on the labels of error presence and error class.
3.	It is shown that the data description is helpful for predicting RE types and locating RE bugs.
Weaknesses:
The proposed extended IPA-GNN has two key modifications: (1) it takes data descriptions as input. (2) it can model the exception handling. My concerns are: (1) The data descriptions are only available for competitive programs, and may not available for other programs. This limits the usage scenario of the approach. (2) I am worrying that most competitive programs tend to not handle exceptions. How many programs in the proposed dataset have exception handling blocks? If there is not many, the exception handling part of the approach could be insignificant. Also, there is no empirical validation of the exception handling in the experiments.


**Summary Of The Paper:**

This paper tries to address the problem of locating runtime error with neural interpreter model. The contribution of this paper includes: (1) a new dataset for the problem, which consists of competitive programs in multiple languages, input data files, input descriptions, runtime error labels. (2) an improved model based on IPA-GNN.

**Summary Of The Review:**

The paper proposes a new dataset as well as a new approach for the problem of static prediction of runtime errors. I appreciate the work of publishing the dataset, but the technical contribution of the proposed approach seems to be thin. The impact of the modeling of exception handling is not fully discussed or evaluated.

---

> ### Author Response · Authors · 2022-11-18
> **Response to Reviewer xTjd**
>
> Thanks for the thoughtful review.
>
> > This paper an interesting problem of locating runtime error with neural interpreter models. The proposed dataset may be useful for the future study in the area.
>
> We're glad to hear you think so! In fact, we are currently (in future work) pursuing runtime error prediction in larger projects, going beyond competitive programming. The dataset from this paper has proved quite useful allowing us to test our ideas in a simpler setting before applying them in the more expensive production job data regime.
>
> > The proposed approach can be used for locating the bug, even though the model is trained with only on the labels of error presence and error class.
>
> We agree that this is one of the most important results in our paper, since it validates the utility of directly modeling exception handling in the model architecture.
>
> >  It is shown that the data description is helpful for predicting RE types and locating RE bugs.
>
> Yes, the data descriptions lead to improved performance across all model types. In competitive programming, the main external resource used by programs is the input, provided on stdin. In other domains, external resources can include flags, input files, RPCs and other forms of network access, or peripheral devices. Sometimes these are naturally accompanied by descriptions (e.g. flags often have help text), which could serve an analogous purpose to the input descriptions in our paper.
>
> > The data descriptions are only available for competitive programs, and may not available for other programs.
>
> We note the validity of your concern that the external resource descriptions we employ are specific to competitive programs. In ongoing work we are extending our approach to predict failures in larger projects, not just competitive programming submissions. There, similar signals to the resource description include command line args/flags and issue tracker text. We acknowledge that the amount of information and format of the information differs in these settings, and that the approach therefore requires modification to apply in this more production-oriented environment. Our experience with this project has been helpful in shaping how we approach the problem.
>
> > I am worrying that most competitive programs tend to not handle exceptions.
>
> You are correct that only a fraction of the competitive programming submissions have explicit exception handling (12615 programs in our dataset). Nevertheless, the component of our model that models exception handling is valuable for all submissions, not just those with explicit exception handling blocks. When the vanilla IPA-GNN makes a confident runtime error prediction, it must continue execution and propagate error information through valid control flow paths until it reaches the exit node. In comparison, when E. IPA-GNN is confident about a runtime error prediction, it can direct probability mass straight to the error node, skipping over code that would not actually be executed, just like how a real interpreter handles exceptions.

---

### Decision · Program_Chairs · 2023-01-20

**Decision:**

Accept: poster

**Justification For Why Not Higher Score:**

Two reviewers felt the results acceptable but not exciting.  The weaknesses they point out do not seem serious to me.  However, the paper does not seem to rise to the level of a spotlight.

**Justification For Why Not Lower Score:**

The innovation in the architecture of control-flow GNNs, and the observation that error localization emerges without training, seems sufficiently significant to warrant publication.

**Metareview: Summary, Strengths And Weaknesses:**

The paper is able to predict runtime errors in a static setting using a form of graph neural network on program points connected by control flow links.  While this has been done before, the paper advances the state of the aret by showing that this can be scaled to larger programs. The paper also shows that this architecture can learn to localize errors without being trained on localization.  Perhaps this is reminiscent of the observation that training a graph network on SAT problems can determine truth assignments when trained only on a satisfiable/unsatisfiable labeling of SAT problems.

A weakness of the paper is that the experiments use home-grown baselines rather than scores from previously published papers on a shared dataset.

**Note From Pc:**

if the above contains the word "oral" or "spotlight" please see: "oral" presentation means -> notable-top-5% and "spotlight" means -> notable-top-25%. As stated in our emails, we are disassociating presentation type from AC recommendations